# Complete Chloroplast Genome of Japanese Larch (*Larix kaempferi*): Insights into Intraspecific Variation with an Isolated Northern Limit Population

**Shufen Chen [1], Wataru Ishizuka [2], Toshihiko Hara [3] and Susumu Goto [1,\*]**

[1] Education and Research Center, The University of Tokyo Forests, Graduate School of Agricultural and Life Sciences, The University of Tokyo, 1-1-1 Yayoi, Bunkyo-ku, Tokyo 113-8657, Japan; suky0414@hotmail.com

[2] Forestry Research Institute, Hokkaido Research Organization, Koushunai, Bibai, Hokkaido 079-0166, Japan; wataru.ishi@gmail.com

[3] Institute of Low Temperature Science, Hokkaido University, Sapporo-city, Hokkaido 060-0819, Japan; t-hara@pop.lowtem.hokudai.ac.jp

\* Correspondence: gotos@uf.a.u-tokyo.ac.jp; Tel.: +81-3-5841-5493

**Abstract:** *Research Highlights:* The complete chloroplast genome for eight individuals of Japanese larch, including from the isolated population at the northern limit of the range (Manokami larch), revealed that Japanese larch forms a monophyletic group, within which Manokami larch can be phylogenetically placed in Japanese larch. We detected intraspecific variation for possible candidate cpDNA markers in Japanese larch. *Background and Objectives:* The natural distribution of Japanese larch is limited to the mountainous range in the central part of Honshu Island, Japan, with an isolated northern limit population (Manokami larch). In this study, we determined the phylogenetic position of Manokami larch within Japanese larch, characterized the chloroplast genome of Japanese larch, detected intraspecific variation, and determined candidate cpDNA markers. *Materials and Methods:* The complete genome sequence was determined for eight individuals, including Manokami larch, in this study. The genetic position of the northern limit population was evaluated using phylogenetic analysis. The chloroplast genome of Japanese larch was characterized by comparison with eight individuals. Furthermore, intraspecific variations were extracted to find candidate cpDNA markers. *Results:* The phylogenetic tree showed that Japanese larch forms a monophyletic group, within which Manokami larch can be phylogenetically placed, based on the complete chloroplast genome, with a bootstrap value of 100%. The value of nucleotide diversity ($\pi$) was calculated at 0.00004, based on SNP sites for Japanese larch, suggesting that sequences had low variation. However, we found three hyper-polymorphic regions within the cpDNA. Finally, we detected 31 intraspecific variations, including 19 single nucleotide polymorphisms, 8 simple sequence repeats, and 4 insertions or deletions. *Conclusions:* Using a distant genotype in a northern limit population (Manokami larch), we detected sufficient intraspecific variation for the possible candidates of cpDNA markers in Japanese larch.

**Keywords:** cpDNA; next generation sequencing; northern limit; nucleotide diversity; phylogeny; In/Del; SNP; SSR; Pinaceae

## 1. Introduction

The chloroplast genome is highly conserved and has a much lower mutation rate than the nuclear genome [1]. Chloroplast DNA (cpDNA) has been widely used to clarify interspecific relationships, and to evaluate the magnitude of intraspecific variation [2,3]. The cpDNA of gymnosperms, particularly of the conifers, is characterized by high levels of intraspecific variation [4,5] and paternal inheritance [6].

A high-resolution chloroplast-specific polymorphic assay would facilitate the analysis of population differentiation and gene flow in gymnosperms [7].

The next-generation sequencing (NGS) technique enables the sequencing of whole chloroplast genomes. The chloroplast genome has a circular molecular structure, with a length ranging from 120 to 160 kbp in most plants. The cpDNA contains a pair of inverted repeats (IRs), a large single-copy region (LSC), and a small single-copy region (SSC) [8]. IRs are a crucial feature of the chloroplast genome in most plants, likely contributing to the maintenance of a conserved arrangement of cpDNA sequences. Previous studies have reported that the length of observed short IRs is roughly consistent among gymnosperm species [9]. The whole chloroplast genome is of significant use for phylogenetic studies [2,3]; Parks et al. [10] presented complete chloroplast genomes for 37 pine species and documented a notable degree of variation at several loci (particularly at *ycf*1 and *ycf*2). Intraspecific variation in whole chloroplast genomes derived from multiple individuals can clarify the phylogenic lineage of target individuals [11]. In particular, single nucleotide polymorphisms (SNPs) have been efficiently used in the fields of phylogeography and conservation biology [12].

Japanese larch (*Larix kaempferi* (Lamb.) Carr.) is a deciduous coniferous tree species endemic to Japan, and integral to the country's forestry efforts. The natural distribution of Japanese larch is limited to the mountainous range in the central part of Honshu Island, Japan [13]. An isolated population with ten mature trees was discovered at Manokami (hereafter Manokami larch), in the Zao Mountains in 1932 [14], extending the known northern limit of the species. Manokami larch was initially believed to be *Larix gmelinii* var. *japonica,* based on the morphological traits. An analysis of partial cpDNA sequences, and random amplified polymorphic DNA (RAPD) analysis, indicated that the Manokami larch population was actually Japanese larch [15]. However, the phylogenic position of Manokami larch, with relation to Japanese larch, has not yet been sufficiently defined [16,17].

The complete chloroplast genome of the genus *Larix* has been reported for several species [11,18,19], and the complete chloroplast genome of Japanese larch introduced in Korea, was reported by Kim et al. [20]. However, the intraspecific variations of Japanese larch have not yet been examined based on the complete chloroplast genome.

In this study, we identified the complete chloroplast genome for eight individuals of Japanese larch, including from the isolated population at the northern limit of the range (Manokami larch), to (1) determine the phylogenetic position of Manokami larch within Japanese larch, (2) characterize the chloroplast genome of Japanese larch, with included chloroplast data from Manokami larch, and (3) detect intraspecific variation and determine candidate cpDNA markers.

## 2. Materials and Methods

Eight individuals of Japanese larch were used in this study. Five individuals (*Lk_Ho1*, *Lk_Ho2*, *Lk_Ho3*, *Lk_Ho4*, and *Sorachi 3*) were collected from test plantations or the Arboretum Garden at the Forestry Research Institute, Hokkaido Research Organization (HRO). *Sorachi 3* was selected specifically due to its superior growth in a larch-breeding program in Hokkaido. Two individuals (*Lk_Ka1* and *Lk_Ka2*), were collected from the open-pollinated progeny of artificial plantations of Japanese larch, at the southern edge of Sakhalin Island, Russia. The detailed location of the seed collection of *Lk_Ka2* was described in a previous study [11]. These seven individuals were originally derived from the eastern region of Nagano, in the central part of Honshu Island in Japan, near the center of distribution for this species. We added one grafted tree (*Manokami 15*), as an isolated germplasm of Manokami larch. The ortet of this tree was conserved in situ with the label "No.15" on Mt. Manokami in the Zao Mountains.

Fresh leaves from all eight individuals were collected in 2016 March for *Lk_Ka2*, June for *Lk_Ho2*, *Lk_Ho3*, *Lk_Ho4*, *Sorachi3*, 2017 March for *Lk_Ka1*, June for *Lk_Ho1*, and 2018 July for *Manokami15*. Leaf sampling, isolation of purified intact chloroplasts, and extraction of high-concentrate chloroplast DNA were performed as previously described [11]. Briefly, we used a saline Percoll (GE Healthcare,

Uppsala, Sweden) gradient for chloroplast isolation and the DNeasy Plant Mini kit (QIAGEN, Hilden, Germany) for DNA extraction.

The cpDNA sequence reads were obtained using the Illumina platform. CLC Genomics Workbench 9.5.3 software (CLC bio, Aarhus, Denmark) was used for genetic analysis. After trimming low-quality sequences from the reads, bulked reads for all eight individuals were used to determine the draft consensus sequence for *L. kaempferi*. Reference mapping to *L. gmelinii* var. *japonica* (LC228570; [11]) was performed with parameter settings of mismatch cost 3, In/Del cost 3, length fraction 0.9, and similarity fraction 0.9. The complete chloroplast genome for each sample was then determined by mapping reads for each sample to our consensus sequence, using the same parameter settings as described above. The initial annotation of the chloroplast genome was performed using DOGMA [21]. Prediction of tRNA genes was performed using tRNAscan (http://lowelab.ucsc.edu/tRNAscan-SE). The annotation was finalized, with reference to that of *L. gmelinii* var. *japonica* (LC228570). To estimate the pseudogenes of *ndh* (subunits of an NADH dehydrogenase), we referred to the *Pinus thunbergii* chloroplast genome (NC_001631; [9]). REPuter [22] was used to confirm repetitive sequences in the chloroplast genome, (i.e., tandem repeats, duplicated genes, and IR regions). Finally, the gene map of the circular chloroplast genome of Japanese larch was drawn using OrganellarGenomeDRAW [23].

A phylogenetic tree was constructed based on the chloroplast genome sequences of the eight individuals identified in this study, and of MF990369 in the NCBI database (https://www.ncbi.nlm.nih.gov/) for Japanese larch, as well as five reference sequences of related *Larix* species derived from the database: LC228570 (*L. gmelinii* var. *japonica*), MF990370 (*L. gmelinii* var. *olgensis*), NC_016058 (*L. decidua*), KX880508 (*L. potaninii*), and NC_036811 (*L. sibirica*). The alignment of these fourteen chloroplast genome sequences was performed in MAFFT [24], and the final alignment was checked using CLC Genomics Workbench 9.5.3. A phylogenetic tree was constructed by MEGA X [25], based on maximum likelihood (ML) methods. A total of 1000 bootstrap replicates were applied to evaluate the branch supports.

The SNP data from the eight Japanese larch individuals was used for subsequent analyses. Haplotype networks have been demonstrated to show alternative genealogical relationships at the intraspecific population level, with low divergence [26]. We estimated haplotype networks for the chloroplast data using the software Network 10 (https://www.fluxus-engineering.com/sharenet.htm). Nucleotide diversity can be used as an inference parameter for evolutionary and demographic forces [12]; here, nucleotide diversity was calculated using DnaSP v6 software [27] and estimated as $\pi$. Divergent regions of the chloroplast genomes were identified according to the variation in $\pi$, by sliding window analysis, with a 500 bp step size and 10,000 bp window length.

Tandem repeats were identified using the Tandem Repeats Finder website [28]. In addition, simple sequence repeats (SSRs) were identified by MISA-web (https://webblast.ipk-gatersleben.de/misa/) [29], with minimum repetition numbers of 10 for mononucleotides, 6 for dinucleotides, and 5 for trinucleotides, tetranucleotides, pentanucleotides, and hexanucleotides each.

## 3. Results

### 3.1. Phylogenetic Analysis

The phylogenetic tree showed that Japanese larch forms a monophyletic group, within which Manokami larch can be phylogenetically placed based on the complete chloroplast genome, with a bootstrap value of 100% (Figure 1). Japanese larch is genetically close to *L. decidua* and *L. gmelinii*, but distant from *L. sibirica* and *L. potaninii* (Figure 1). The haplotype network among the eight sampled individuals revealed that Manokami larch was genetically distinct from other Japanese larches (Figure 2).

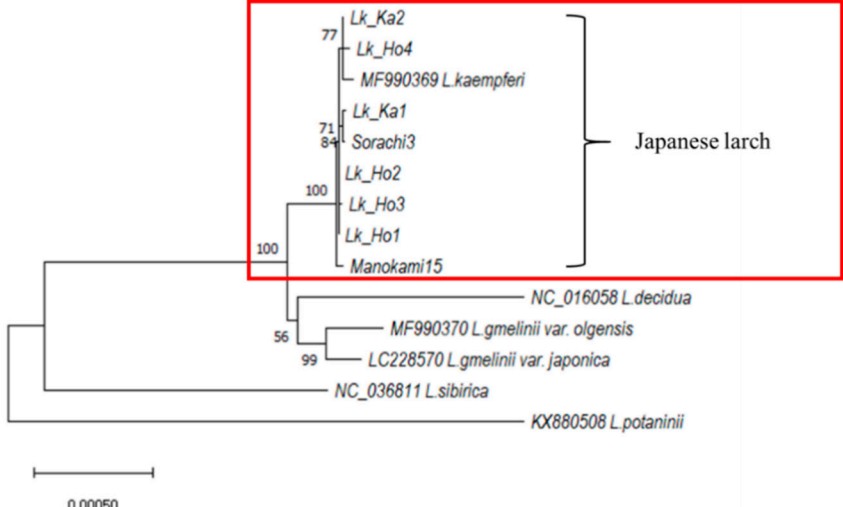

**Figure 1.** The maximum likelihood (ML) phylogenetic tree based on 14 chloroplast genomes of *Larix* species. The red square represents Japanese larch.

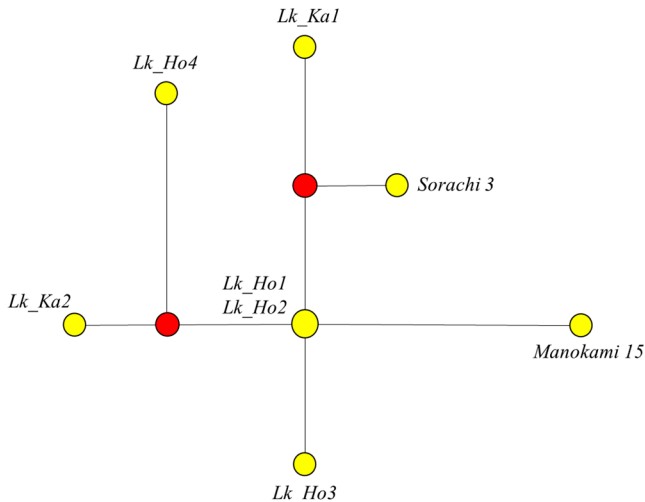

**Figure 2.** Haplotype network based on SNP sites within eight individuals of Japanese larch. The red dot represents missing haplotypes.

### 3.2. Characteristics of the Japanese Larch Chloroplast Genome

Japanese larch circular chloroplast genomes were characterized in the range of 122,394–122,409 bp with accession numbers from the DNA Data Bank of Japan (DDBJ) from LC574969 to LC574976. The gene type, number, and order were identical among the Japanese larch chloroplast genomes used in this study. *Lk_Ho1* (LC574969) was used as a representative of Japanese larch; Figure 3 illustrates the physical mapping of its chloroplast genome, which contained a pair of IRs (436 bp each) separated by large single copy (LSC), and small single copy (SSC) regions, of 65,398 bp and 56,136 bp, respectively. The *trnI-CAU* gene was duplicated within inverted repeats, the *trnS-GCU* and *psbI* genes were duplicated as another inverted repeat of 457 bp in the LSC region (Figure 3), and two *trnT-GGU* genes were dispersed in the LSC and SSC regions, respectively. A total of 119 genes were identified (Table S1), including 72 protein genes, 35 transfer RNA genes, 4 ribosomal RNA genes, and 8 pseudogenes. Thirteen genes contained an intron, including *trnA-UGC*, *trnG-UCC*, *trnI-GAU*, *trnK-UUU*, *trnL-UAA*, *trnV-UAC*, *rpoC1*, *rps12*, *rpl2*, *rpl16*, *petB*, *petD*, and *atpF*. In addition, *ycf3* contained two introns. Furthermore, *rps12* was a trans-splicing gene with 5′ end and 3′ end exons, located in the LSC region and the SSC region, respectively. The G+C content of the complete chloroplast genome of Japanese larch was 38.7%.

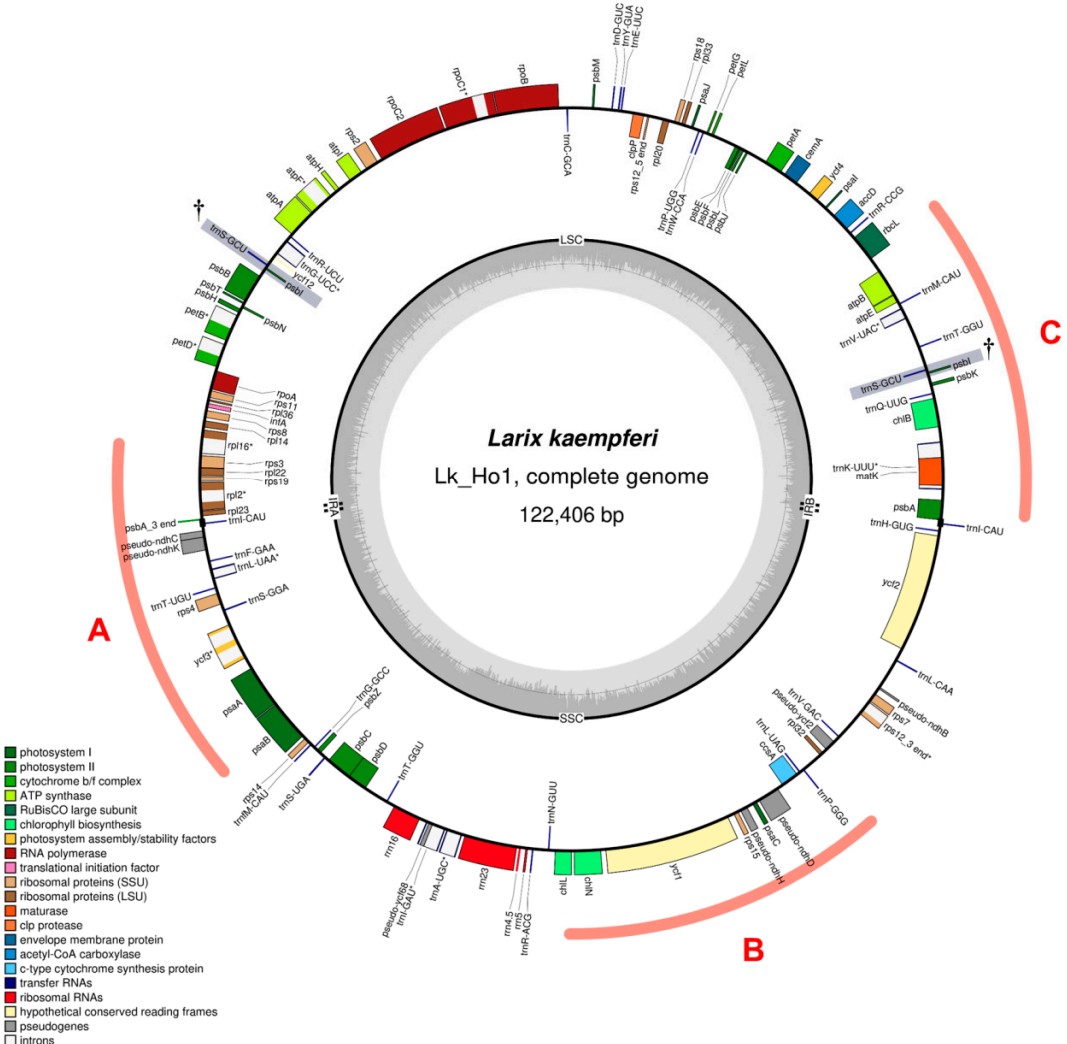

**Figure 3.** A gene map of Japanese larch chloroplast genome (accession number: LC574969). Genes shown outside and inside the circle are transcribed clockwise, and transcribed counterclockwise, respectively. Genes were colour-coded to distinguish different functional groups. The dark and light gray inner circle indicates the GC and AT content of the chloroplast genome, respectively. "†" represents the location of a longer inverted repeat. A, B and C represent hotspot of variation.

### 3.3. Nucleotide Diversity Analysis

The value of nucleotide diversity (π) was calculated at 0.00004, based on SNP sites for Japanese larch, suggesting that sequences had low variation. As shown in Figure 4, there were three divergent regions (A, B, and C) in Japanese larch. Two regions (A, C), which were roughly in the range of the *rpl16* gene *psaB* and the *rbcL* gene *psbA*, respectively, were classified as moderately variable (π > 0.00004); these regions contained variant sites in *psaB*, *rpl16*, ψ*ndhK*, *atpB*, *psbK*, and *matK*, and three intergenic spacers (between the *rpl23* and *psbA*-partial genes, between the *trnS-GGA* and *ycf3* genes, and between the *trnS-GCU* and *trnT-GGU* genes). The B region (roughly from *chlL* to *rpl32*, π > 0.0001) was identified as a hypervariable region, in which mutation occurred twice in the ψ*ndhD* and the *ycf1*.

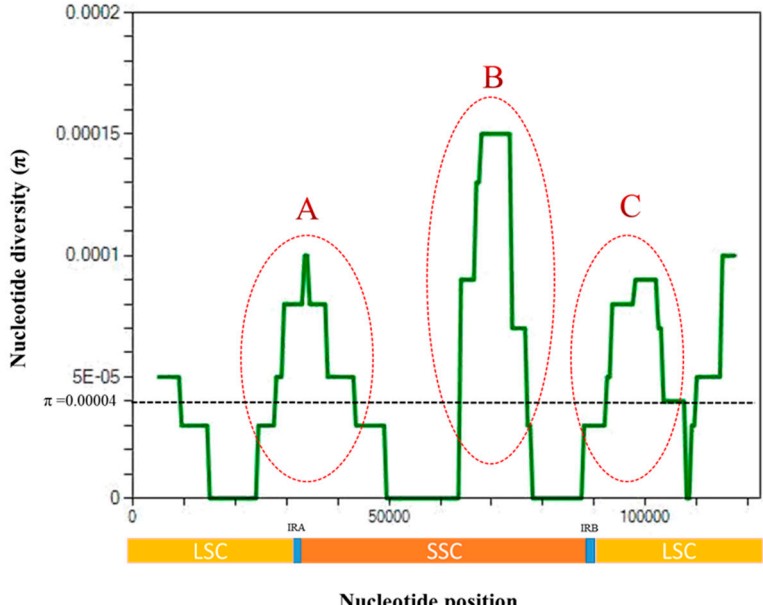

**Figure 4.** The nucleotide diversity (π) of Japanese larch chloroplast genomes, based on sliding window analysis.

### 3.4. Repeat Sequence Analysis of the Japanese Larch Chloroplast Genome

Tandem repeats were detected in approximately 25 sites in the Japanese larch chloroplast genomes. Repeated lengths of tandem repeats varied from 12 to 117 bp, and 64% of all tandem repeats occurred in *ycf1*, which belongs to a protein-coding region containing 76% of all detected tandem repeats. Nineteen SSR motifs were detected in the Japanese larch chloroplast genome. The majority of the detected SSR motifs were mononucleotide motifs, of which the SSR motif of mononucleotide T was the most frequent, followed by mononucleotide A and mononucleotide G. With the exception of three SSR motifs of dinucleotide AT, no other multiple nucleotide motifs were detected in the Japanese larch chloroplast genome. Furthermore, most (77.8%) of the detected SSR motifs were found in the intergenic region, followed by introns (16.7%), and protein-coding genes (5.5%). Eight cpSSR variants out of nineteen cpSSR motifs were detected in the intergenic region of the chloroplast genome of Japanese larch.

### 3.5. Genetic Variation among Japanese Larch Chloroplast Genomes

Among the eight individuals sequenced in this study, 31 variants (including 19 SNPs, 8 SSRs, and 4 In/Dels) were detected. For SNP variants, six and thirteen SNPs were identified in the intergenic spacer (IGS) and coding sequence (CDS) regions, respectively. These were detected in the ψ*ndhK* (one SNP), ψ*ndhD* (two SNPs), and protein-coding *ycf1* (two SNPs), all of which belong to the SSC region. Six SSR variants were identified in the IGS, whereas two SSR variants were detected in the CDS region. Four In/Del variants were identified in the ψ*ndhK* gene (one In/Del variant) and the *ycf1* gene (three In/Del variants), belonging to CDS region. (Table 1)

**Table 1.** Genetic variation among chloroplast genomes of Japanese larch. CDS: coding sequence; IGS: intergenic spacer; In/Del: insertion or deletion; SNP: single nucleotide polymorphism; SSR: simple sequence repeat.

| Sequence | N | In/Del | | SNP | | SSR | |
|---|---|---|---|---|---|---|---|
| | | CDS | IGS | CDS | IGS | CDS | IGS |
| *Lk_Ho1*(Reference) | — | — | — | — | — | — | — |
| *Lk_Ho2* | 2 | | | | | | 2 |
| *Lk_Ho3* | 3 | 1 | | 1 | 1 | | |
| *Lk_Ho4* | 8 | | | 4 | 2 | | 2 |
| *Lk_Ka1* | 4 | | | 3 | 1 | | |
| *Lk_Ka2* | 5 | 1 | | 3 | | | 1 |
| *Sorachi3* | 3 | | | 3 | | | |
| *Manokami15* | 12 | 2 | | 2 | 3 | 1 | 4 |
| Whole sequences | 31 | 4 | | 19 | | 8 | |

## 4. Discussion

The phylogenetic position of Manokami larch, has been discussed by several researchers [14,16,17]. This study clearly indicates that Manokami larch should be phylogenetically categorized into Japanese larch, with a bootstrap value of 100% (Figure 1). Our findings support the assertion by Shiraishi et al. [15] that Manokami larch must be a Japanese larch. Manokami larch is located far from other Japanese larches (Figure 2); genetically divergent genotypes, such as that of Manokami larch, could be used to efficiently detect intraspecific variation in Japanese larch.

The chloroplast genomes of Japanese larch obtained from this study were similar in size and gene order to those of *L. gmelinii* [11], *L. sibirica* [19], *L. decidua* [30], and *L. potaninii* [18]. The chloroplast structure types were classified in Pinaceae according to their alignment order and the orientation of the F1 (fragment flanked by *trnG-UCC* and *trnE-UUC*), F2 (fragment flanked by *clpP* and *trnT-GGU*), T1 (type 1 Pinaceae-specific repeat containing *trnS-GCU* and *psbI*), and T2 (type 2 Pinaceae-specific repeats in intergenic spacers) fragments in the LSC region, which can produce eight different cpDNA forms, including A, B, C, D, E, F, and G forms [30]. The chloroplast DNA form used in this study was classified into the C form, the same form identified for *L. gmelinii*, *L. decidua*, *L. griffithiana*, and *Pinus elliottii* [11,30] based on the alignment order and orientation of T1, T2, −F1 (reverse strand), T2, +F2 (forward strand), and T1. Due to this T1 repeat, there were longer inverted repeats (457 bp) in the LSC region than two IRs (436 bp) in Japanese larch. Extremely shortened IRs, with another pair of inverted repeats, is regarded as a common feature in Pinaceae [30,31].

In this study, three hotspots of variation were detected throughout the entire chloroplast genome (Figures 3 and 4). The *ycf1* and *ψndhD* were included in the hypervariable region (region B), and the *ψndhK* was included in the moderately variable region (region A). Three In/Del variants occurred in the *ycf1*, and previous research has reported insertions or deletions in the *ycf1* of *L. gmelinii* [11]. Although it was considered a possibility that the *ycf1* might be a nonfunctional pseudogene, another study [32] indicated that *ycf1* is a functional gene, and encodes a product essential for cell survival. Dong et al. [33] revealed that the divergence of the *ycf1* was obvious in gymnosperms. Additionally, Firetti et al. [34] indicated that the *ycf1* was more divergent than the non-coding regions in the genus *Anemopaegma*. Regarding pseudogenes, eleven *ndh* genes (*ndhA—K*) have been identified in the cpDNA sequences of photosynthetic land plants [9,35,36]. In our study, five *ψndh* genes were found only in Japanese larch, of which *ψndhD* (two SNPs) and *ψndhK* (one SNP, one SSR, one In/Del variant) belonged to the region of frequent variation; these genes did not, however, exhibit a function consistent with other *Pinus* and *Larix* species [9,11].

Repeat sequences may play an important role in chloroplast genome arrangement and sequence divergence. In particular, tandem repeats may induce In/Dels [37,38]. In this study, tandem repeats

were primarily identified in the *ycf1*. Tandem repeats were also located in the *ycf1* of other conifers, such as *Cryptomeria japonica* [39] and *L. gmelinii* [11].

Among the eight Japanese larch individuals, we detected 31 variants (19 SNPs, 8 SSRs, and 4 In/Dels) located in *psaB*, *ψndhD*, *ψndhK*, *psbE*, *psbK*, *rpoC1*, *rpoC2*, the intron of *rpl16*, *matK*, *atpB*, *ycf1*, six intergenic spacers (between the *rpl23* and *psbA*-partial genes, between the *trnS-GGA* and *ycf3* genes, between the *trnS-GCU* and *trnT-GGU* genes, between the *clpP* and *trnE-UUC* genes twice, between the *psbE* and *petL* genes) with SNPs, the intron of *atpF*, *ψndhK*, six intergenic spacers (between the *trnC-GCA* and *rpoB* genes, between the *ycf1* and *rps15* genes, between the *trnL-CAA* and *ycf2* genes, between the *ψycf2* and *trnV-GAC* genes, between the *trnT-GGU* and *trnV-UAC* genes twice) with SSRs, *ψndhK* and *ycf1* with In/Dels that could prove useful for providing candidate cpDNA markers. Chloroplast simple sequence repeat (cpSSR) markers often contain highly polymorphic variations within a population of conifers (see [7]), although Zhang et al. [40] found only three polymorphic cpSSR markers among 11 candidate markers in Japanese larch. We identified 19 SSR motifs within the chloroplast genome of Japanese larch, preferentially within the intergenic space, and only 8 SSR motifs occurred among 19 SSR motifs in the intergenic region of the chloroplast genome. These results lay a foundation for the development of cpDNA markers for Japanese larch.

## 5. Conclusions

The complete chloroplast genome of Japanese larch (122,398–122,409 bp) was obtained using next-generation sequencing technology. The comparison of whole chloroplast genomes clearly indicated that the isolated population, forming the northern limit of the species' range (Manokami larch), should be placed phylogenetically within Japanese larch. The Manokami larch was found to be genetically different from other Japanese larches, indicating that sufficient genetic variation should be detected within the samples used in this study. Based on an analysis of intraspecific variation, 31 variants were detected, including 19 SNPs, 8 SSRs, and 4 In/Dels, all of which can be applied for the development of cpDNA markers. These variations should be useful for paternity analysis and population genetics analysis of Japanese larch in future studies.

**Supplementary Materials:** The following are available online at http://www.mdpi.com/1999-4907/11/8/884/s1, Table S1: List of estimated chloroplast genes of Japanese larch. Genes with * have intron(s). Duplicated genes are shown in parenthesis.

**Author Contributions:** Conceptualization, W.I. and T.H.; Founding acquisition, T.H.; Software, W.I. and S.C.; visualization, W.I; Validation, W.I, S.C., T.H. and S.G.; Formal analysis, S.C. and W.I.; data curation, S.C. and W.I.; Writing—original draft preparation, S.C., S.G.; Writing—review and editing, S.C., W.I., T.H. and S.G. All authors have read and agreed to the published version of the manuscript.

**Funding:** This study was partly supported by the Japan Society for the Promotion of Science [15K18715] and the Grant for Joint Research Program of the Institute of Low Temperature Science, Hokkaido University.

**Acknowledgments:** We would like to thank K. Ono for the help of laboratory work.

**Conflicts of Interest:** The authors declare no conflict of interest.

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
