# Peer review of "Complete Chloroplast Genome of Japanese Larch (Larix kaempferi): Insights into Intraspecific Variation with an Isolated Northern Limit Population"

_forests, doi:10.3390/f11080884_

Round 1

Reviewer 1 Report

I think this manuscript is well organized, and provides useful information about intraspecific cpDNA variation and phylogenetic insight of the isolated population in L. kaempferi

This manuscript provided further detailed information on intra-species cpDNA variation of L. kaempferi comparing with previously reported research by sequencing cpDNA genomes of multiple individuals. The results provides various haplotypes by cpDNA variation among the intra-species individuals. DNA variation within species is a interesting subject in the population genetics sector. So, this information will be useful in developing markers not only to evaluate genetic diversity within L. kaempferi population but also to analyze parentage and mating system. In addition, the phylogenetic results including the individual from the marginal population give useful information for the effective management as a same species. Meanwhile, the results of this manuscript are well suited to the purpose of the research, and were logically demonstrated by the experimental method and materials.

Author Response

Comments from Reviewer#1

I think this manuscript is well organized, and provides useful information about intraspecific cpDNA variation and phylogenetic insight of the isolated population in L. kaempferi

This manuscript provided further detailed information on intra-species cpDNA variation of L. kaempferi comparing with previously reported research by sequencing cpDNA genomes of multiple individuals. The results provides various haplotypes by cpDNA variation among the intra-species individuals. DNA variation within species is a interesting subject in the population genetics sector. So, this information will be useful in developing markers not only to evaluate genetic diversity within L. kaempferi population but also to analyze parentage and mating system. In addition, the phylogenetic results including the individual from the marginal population give useful information for the effective management as a same species. Meanwhile, the results of this manuscript are well suited to the purpose of the research, and were logically demonstrated by the experimental method and materials.

Answer: Thank you so much for taking your time to review this manuscript. Thank you so much for your help. We are going to develop cpDNA markers from findings obtained in this study in the next study.

Reviewer 2 Report

The review of manuscript titled “Complete chloroplast genome of Japanese larch (Larix kaempferi): insights into intraspecific variation with an isolated northern limit population" by Shufen Chen, Wataru Ishizuka, Toshihiko Hara, Susumu GOTO

 The manuscript by Susumu GOTO (corresponding author) and others aimed at determining the phylogenetic position of the Manokami larch population within Japanese larch (Larix kaempferi). This work characterizes the chloroplast genome and cpDNA markers of this population. The subject and the results of the presented investigation are restricted to the small population of Manokami larch, which is genetically different in terms of Japanese larch. The results of this study could be useful for molecular markers developing to distinguish Manokami larch from Japanese larch.The results of the experiment are interesting from the point of view of the conservation and management of genetic variation of Japanese larch. The manuscript is well-written; however, it contains several deficiencies that require attention and should be corrected.

Abstract and Introduction

The Abstract contains essential results and the most important conclusions. The Introduction section contains the most important and actual state of knowledge in the manuscript subject. The authors present the important aspects characterizing the genetic features of cpDNA. They also show the natural distribution of Larix kaempferi and phylogenetic traits of Manokami larch.

The Materials and Methods section presents necessary information connected with biological material, cpDNA sequence read methods, and sequence analysis.

The results are correctly described and based on data obtained.

The authors mention that they detected sufficient intraspecific variation for cpDNA markers identification in Japanese larch (including Manokami larch) (for example, in Abstract (Line 19-20; 34-36), in Discussion (Line 270-271; 276-277), in Conclusion (Lines 286-287). In my opinion, it should be described in more detail. Why is it not given details on these markers - sequences, genes, or gene fragments?

Editing errors:

Line 23: phylogenic change to phylogenetic

Lines 145, 149, 170, 187: Figureure change to Figure

References:

Line 318: Abies mariesii change to Abies mariesii (italics)

Line 322: Pinaceae change to Pinaceae (italics)

Line 341: Larix change to Larix (italics)

Line 378: Pinaceae change to Pinaceae (italics)

Line 398: Angiosperm change to Angiosperm (italics)

Line 401: Geraniaceae change to Geraniaceae (italics)

Author Response

#Comments from Reviewer#2

The review of manuscript titled “Complete chloroplast genome of Japanese larch (Larix kaempferi): insights into intraspecific variation with an isolated northern limit population" by Shufen Chen, Wataru Ishizuka, Toshihiko Hara, Susumu GOTO

 The manuscript by Susumu GOTO (corresponding author) and others aimed at determining the phylogenetic position of the Manokami larch population within Japanese larch (Larix kaempferi). This work characterizes the chloroplast genome and cpDNA markers of this population. The subject and the results of the presented investigation are restricted to the small population of Manokami larch, which is genetically different in terms of Japanese larch. The results of this study could be useful for molecular markers developing to distinguish Manokami larch from Japanese larch. The results of the experiment are interesting from the point of view of the conservation and management of genetic variation of Japanese larch. The manuscript is well-written; however, it contains several deficiencies that require attention and should be corrected.

Answer: Thank you for your helpful comments. We significantly revised the manuscript as following to your comments.

Abstract and Introduction

The Abstract contains essential results and the most important conclusions. The Introduction section contains the most important and actual state of knowledge in the manuscript subject. The authors present the important aspects characterizing the genetic features of cpDNA. They also show the natural distribution of Larix kaempferi and phylogenetic traits of Manokami larch.

Answer: Thank you so much for taking your time to review this manuscript. We revised the Background and Objectives: of the abstract as follows; ”The natural distribution of Japanese larch is limited to the mountainous range in the central part of Honshu Island, Japan with an isolated northern limit population (Manokami larch). In this study, we determined the phylogenetic position of Manokami larch within Japanese larch, and characterized the chloroplast genome of Japanese larch, and detected intraspecific variation and determine candidate cpDNA markers.”

The Materials and Methods section presents necessary information connected with biological material, cpDNA sequence read methods, and sequence analysis.

Answer: We added the information as following to the comments.

The results are correctly described and based on data obtained.

Answer: Thank you for your suggestions.

The authors mention that they detected sufficient intraspecific variation for cpDNA markers identification in Japanese larch (including Manokami larch) (for example, in Abstract (Line 19-20; 34-36), in Discussion (Line 270-271; 276-277), in Conclusion (Lines 286-287). In my opinion, it should be described in more detail. Why is it not given details on these markers - sequences, genes, or gene fragments?

Answer: Among the eight Japanese larch individuals, we detected 31 variants (19 SNPs, 8 SSRs, and 4 In/Dels). In this study, we did not develop cpDNA markers. However, we are planning to develop cpDNA markers (especially cpSNPs and cpSSRs) and will use these markers to conduct the paternity analysis of offspring derived from the seed orchard in the next paper.

Editing errors:

Line 23: phylogenic change to phylogenetic

Answer: We revised the term as following to the comment.

Lines 145, 149, 170, 187: Figureure change to Figure

Answer: We revised the terms as following to the comment.

References:

Line 318: Abies mariesii change to Abies mariesii (italics)

Line 322: Pinaceae change to Pinaceae (italics)

Line 341: Larix change to Larix (italics)

Answer: Thank you for your suggestions. We italicized these terms (Line 318, 341). Although we checked the PDF files of the articles, it was represented in normal font in the original paper (Line 322).

Line 378: Pinaceae change to Pinaceae (italics)

Line 398: Angiosperm change to Angiosperm (italics)

Line 401: Geraniaceae change to Geraniaceae (italics)

Answer: Thank you for your comments. Although we checked the PDF files of the articles, they were represented in normal font in the original paper (Line 378, 398, and 401).

Reviewer 3 Report

Chen and coworkers report the complete chloroplast genome of Japanese larch obtained via NGS. The phylogenetic analysis shows that Japanese larch forms a monophyletic group. Manokami larch was found to be genetically different from other Japanese larches, indicating a great genetic variation within the studied samples. Moreover, authors detected intraspecific variation for possible candidate cpDNA markers in Japanese larch.

Manuscript is generally well written, results are clearly presented and are interesting by themselves and for their possible future applications.

Just little things: the whole sequence was submitted to the DNA data bank of Japan (DDBJ), if authors have not yet received the accession number, they can report the temporary one provided; an accurate reading of the article is required, there are several little errors in the text (figureure instead of the figure i.e.).

Best regards 

Author Response

Comments from Reviewer#3

Comments and Suggestions for Authors

Chen and coworkers report the complete chloroplast genome of Japanese larch obtained via NGS. The phylogenetic analysis shows that Japanese larch forms a monophyletic group. Manokami larch was found to be genetically different from other Japanese larches, indicating a great genetic variation within the studied samples. Moreover, authors detected intraspecific variation for possible candidate cpDNA markers in Japanese larch.

Manuscript is generally well written, results are clearly presented and are interesting by themselves and for their possible future applications.

Just little things: the whole sequence was submitted to the DNA data bank of Japan (DDBJ), if authors have not yet received the accession number, they can report the temporary one provided; an accurate reading of the article is required, there are several little errors in the text (figureure instead of the figure i.e.).

Answer: Thank you so much for taking your time to review this manuscript and giving us comments. We already revised editing errors in the text. We feel very sorry that we cannot provide the temporary accession number for you as it is still in the process of checking submitted data and we haven’t gotten reply from DDBJ yet, but, we think we may get reply from DDBJ soon. Now, we would like to share the files of DDBJ registration to you. Please review the files attached as appendix, we hope it is useful. We appreciate it that you gave us your help.
